# Recent Advances of Marine Natural Indole Products in Chemical and Biological Aspects

**DOI:** 10.3390/molecules28052204

**Published:** 2023-02-27

**Authors:** Haoyi Sun, Kangping Sun, Jingyong Sun

**Affiliations:** 1School of Parmacy and Pharmaceutical Sciences, Institute of Materia Medical, Shandong First Medical University & Shandong Academy of Medical Sciences, Jinan 250117, China; 2NHC Key Laboratory of Biotechnology Drugs, Shandong Academy of Medical Sciences, Jinan 250117, China; 3Key Laboratory for Rare & Uncommon Discases of Shandong Province, Jinan 250117, China

**Keywords:** marine natural products, indole, pharmacological activity, total synthesis, lead compound

## Abstract

The ocean has always been one of the important sources of natural products. In recent years, many natural products with different structures and biological activities have been obtained, and their value has been clearly recognized. Researchers have been deeply engaged in the field of separation and extraction, derivative synthesis, structural studies, biological evaluation, and other fields of research for marine natural products. Thus, a series of marine indole natural products which have structural and biological prospect have caught our eyes. In this review, we summarize some of these marine indole natural products with relatively good pharmacological activity and research value, and discuss issues concerning chemistry, pharmacological activity, biological evaluation, and synthesis, including monomeric indoles, indole peptides, bis-indoles, and annelated indoles. Most of the compounds have cytotoxic, antiviral, antifungal, or anti-inflammatory activities.

## 1. Introduction

The marine environment has been explored for the purpose of searching for new bioactive compounds over the past 50 years, becoming an important and rich source of potent molecules and lead compounds. Alkaloids, which constitute one of the largest classes of natural products, are synthesized by terrestrial and marine organisms on all evolutionary levels, usually present in organisms as mixtures consisting of several major and a few minor compounds. These compounds have the same biosynthetic source, and the differences only appear in functional groups.

This class of compounds has apparently evolved as a defense mechanism against predators, and as a result, alkaloids are often highly potent and toxic molecules [1]. Marine invertebrates have proven to be an outstanding source of active molecules, one of the most promising being indole alkaloids.

Indole alkaloids, their activity, synthesis, and potential use in medicine have already been reviewed in several articles [2]. In this review, we provide information on current and potential pharmaceuticals including small molecule marine indole alkaloids, their biological properties, and structure-activity relationship studies.

## 2. Monomeric Indoles and Annelated Indoles

Jiang et al. isolated two new alkaloids, phidianidines A (**1**) and B (**2**) (Figure 1), from the marine opisthobranch mollusk *Phidiana militaris*. They are the first natural products with 1,2,4-oxadiazole ring system, which show strong antitumor activity against C6 and Hela cells, with IC_50_ values in the nanomolar range [3], and they were synthesized for the first time by Guo et al. (Figure 1). Compound **1** and synthetic analogues showed immunosuppressive properties [4]. In addition, compound **1** was also shown to be a selective inhibitor of the dopamine transporter and a selective, potent ligand and partial agonist of the μ-opioidreceptor (versus δ- and κ-opioid receptors) [5]. In addition, by using virtual screening and experimental method, compound **1** was considered as a new antagonist of CXCR4 which is a chemokine receptor associated with several diseases such as HIV, rheumatoid arthritis, and cancer [6]. Compound **1** is also a potent natural antifoulant and its structure can be tuned to generate simpler and improved synthetic analogues [7].

Five new indole-terpenoids named penerpenes E–I (**3**–**7**) were isolated from the marine-derived fungus *Penicillium* sp. *KFD28* from a bivalve mollusk, *Meretrix lusoria*, among them, compounds **3**, **4,** and **6** exhibited inhibitory activity against PTP1B with IC_50_ values of 14, 27, and 24 μM. Compound **3** also inhibited PTP sigma with an IC_50_ of 38 μM, while compound **6** inhibited TCPTP with IC_50_ values of 35 μM, respectively [8].

In 2020, two new compounds named epipaxilline **8** and penerpene J were isolated from the marine-derived fungus *Penicillium* sp. *KFD28* by Chen et al. Compounds **8** and **9** showed inhibitory activities against PTP1B with IC_50_ values of 31.5 and 9.5 μM, respectively, and compound **9** also showed inhibitory activities against TCPTP with IC_50_ value of 14.7 μM [9].

With the aid of genomic analysis, eight indole-diketopiperazines, including three new compounds, spirotryprostatin G (**10**) and cyclotryprostatins F and G (**11–12**), were obtained by large-scale cultivation of the marine-derived fungus *Penicillium brasilianum* HBU-136 using rice medium with 1.0% MgCl_2_. In addition, compound **10** proved to be active against HL-60 cell line with the IC_50_ value of 6.0 μM, whereas compounds 11 and 12 are active against McF-7 cell line with the IC_50_ values of 7.6 and 10.8 μM, respectively [10] (Figure 1).

Discodermindole (**13a**) and 6-hydroxydiscodermindole (**13b**), isolated from the Bahama sponge Discodermia polydiscus, and trachycladindoles A-G (**14a**–**g**), isolated from the South Australian sponge *Trachycladus laevispirulifer*, belong to the rather limited class of indole alkaloids containing a 2-aminoimidazoline substituent at position 3. In addition, they all, with the exception of compound **14g**, have one or more bromine atoms in their structure, which is common for many secondary metabolites of marine origin [11,12,13] (Figure 2).

Convoluindole A (**15**) was isolated from the cosmopolitan species *Amathia convoluta Lamouroux* (order Ctenostomata) from the Gulf of Mexico off the coast of Florida, as a pale yellow oil which crystallized after storage at −30 °C overnight, m.p. 61.5–62.5 °C. The molecular formula was determined by high-resolution liquid second ion mass spectrometry to be C_14_H_17_Br_3_N_2_O_2_. In agreement with this assignment, the isotope pattern was characteristic of a tribrominated compound [14].

In 2020, four new indole diterpenoids, ascandinines A-D (**16–19**), were isolated from an Antarctic sponge-derived fungus *Aspergillus candidus HDN15-152* by Zhou et al. Ascandinine A (**16**) possesses an unprecedented 2-oxabicyclo [2.2.2]octan-3-ol motif embedded in a pentacyclic ring system, while ascandinines B–D (**17–19**) represent a rare type of indole diterpenoid featuring the 6/5/5/6/6/6/6-fused ring system. Ascandinine C (**18**) displayed anti-influenza virus A (H1N1) activity with an IC_50_ value of 26 μM, while ascandinine D (**19**) showed cytotoxicity against HL-60 cells with an IC_50_ value of 7.8 μM [15] (Figure 3).

Meridianin A–G (**20a**–**g**) was discovered by Gompel et al. to be an effective inhibitor of various protein kinases, including casein kinase 1, glycogen synthase kinase-3, cyclic nucleotide-dependent kinases, and cyclin dependent kinases. Meridianins can penetrate cells and disrupt the function of kinases necessary for cell division and death. This results in the prevention of cell proliferation and the induction of apoptosis. These findings imply that meridianins represent a potentially useful framework for the development of more powerful and specific protein kinase inhibitors [16]. The synthesis of meridianin C, D, F, and G (**20c**, **d**, **f**, and **g**) (Figure 2) via a one-pot Masuda borylation-Suzuki coupling sequence was recently introduced by Kruppa et al. [17].

From the boreal sponge *Geodia barretti*, which was discovered off the coast of Norway, several compounds were isolated, including barettin (**21**), 8,9-dihydrobarettin (**22**), 6-bromoconicamin (**23**), and a brand-new brominated marine indole. The compounds were evaluated as inhibitors of electric eel acetylcholinesterase. Compounds **21** and **22** displayed significant inhibition of the enzyme, with inhibition constants (Ki) of 29 and 19 µM, respectively, towards acetyl cholinesterase via a reversible noncompetitive mechanism [18].

Indole derivatives including new bromoindoles have been isolated from the South Pacific marine sponges *Rhopaloeides odorabile* and *Hyrtios* sp. Their potential cytotoxic, antioxidant, and phospholipase A_2_ (PLA_2_) inhibiting properties were valued. The new derivative 5,6-dibromo-L-hypaphorine (**24**) isolated from *Hyrtios* sp. revealed a weak bee venom PLA_2_ inhibition (IC_50_ = 0.2 mM) and a significant antioxidant activity with an Oxygen Radical Absorbance Capacity (ORAC) value of 0.22 [19].

A novel indole diterpene known as penicindopene A (**25**) was discovered in the *Penicillium* sp. *YPCMAC1* deep-sea mold. Containing IC_50_ values of 15.2 and 20.5 µmol, respectively, compound **25** was the first instance of an indole diterpene with a 3-hydroxyl-2-indolone moiety, and it showed mild cytotoxicities against the A549 and HeLa cell lines [20].

From the Red Sea sponge *Hyrtios* sp., Youssef et al. (2013) discovered three novel alkaloids, hyrtioerectines D-F (**26a**–**c**). The rare marine alkaloids known as hyrtioerectines D-F (**26a**–**c**) have a C-3/C-3 linkage between the indole and β-carboline moiety of the molecule. Variable antibacterial, free radical scavenging, and cancer growth suppression properties were shown by hyrtioerectines D-F (**26a**–**c**). According to Table 1 [21], compounds **26a** and **b** were more active than compound **26c** (Figure 4).

Fumigatosides E (**27a**) and F (**27b**), two novel alkaloids, were discovered in the deep-sea fungus *Aspergillus fumigatus SCSIO 41012*. Both compound **27a** and compound **27b** demonstrated strong antifungal activity against *Fusarium oxysporum f.* sp. momordicae with MIC values of 1.56 g/mL and 6.25 g/mL, respectively [22,23] (Figure 5).

## 3. Indolyl Peptides

Two N-acylanthranillic acids (**28a,b**), one of which is a new natural product, were isolated as co-metabolites of bacillamides (**29a,b**) and N-acetyltryptamine by Akiyama et al. using metabolome mining in a strain known as Laceyella sacchari. Anthranilic acid and Ac_2_O or propionyl chloride were combined to synthesis compounds **28a** and **28b** (Figure 3), which were then tested for bioactivity and structure. The physicochemical properties of the synthetic **28a** and **28b** were essentially the same as those of the natural products [23].

While **28b**, **29a**, and **29b** were initially discovered from *Bacillaceae* or *Thermoactinomycetaceae*-**28b** from *Bacillus pantothenicus* [24], **29a** from a marine *Bacillus* sp. as an algicide selective to dinoflagellates and raphydophytes, and **29b** from *B. endophyticus* [25]-compound **28a** was isolated for the first time as a natural product (Figure 6).

A series of linearly fused prenylated indole alkaloids was isolated from *Aspergillus versicolor*, a fungus isolated from the mud of the South China Sea. (Figure 7) and of these compounds, asperversiamides A-C and E (**30**–**32** and **34**) each contain a rare anti-bicyclo [2.2.2] diazaoctane ring, and asperversiamide D (**33**) contains the analogous syn-ring (when the C_21_-C_22_ and C_17_-N_13_ bonds are cofacial, the ring is defined as “syn”, and when the C_21_-C_22_ and C_17_-N_13_ bonds are on opposite faces, the ring is considered “anti”) [26,27].

The first linearly fused indole alkaloid discovered with a rare fused-imine-containing pyrrole ring structure is asperversiamide A (**30**). Additionally, molecules **31**, **32** and **33**, **34** are corresponding pairings of C-3 and C-21 epimers. Asperversiamide G (**36**) has a unique Z-geometry of the double bond between C-10 and C-11, while asperversiamide F (**35**) is the C-17 epimer of dihydrocarneamide A (**37**) [28]. Compound **37**, which has an isoprenyl unit at C-3 and is a key precursor of spiro-bicyclo [2.2.2] diazaoctane type indole alkaloids, is based on the biosynthesis route (**31–32**). By additional modification, co-isolated deoxybrevianamide E (**38**) might be used as a precursor to a number of structurally similar prenylated indole alkaloids [29].

## 4. Bis-Indole Alkaloids

Due to its strong biological activity and novel structural features, there is a strong interest in some bis-indole secondary metabolites containing spacer units derived from imidazole or piperazine, like hamacanthin A (**39**) and B (**40**) [30]. These two compounds, discovered by Gunasekera and his colleagues [31], are two isomeric bis-indole alkaloids isolated from the deep-sea species *Hamacantha* sp. Compound **39** is a 3,6-bis-indole derivative and is similar to dragmacidins, but compound **40** is 3,5-isomer, whose structure is rare among these alkaloids. Since these alkaloids are relatively rare in nature and difficult to extract, the importance of finding a method for total synthesis is apparent. In 2005, Takashi et al. reported the method of total synthesis of marine bisindole alkaloids, compounds **39** and **40** (Figure 8). Therein, they describe the total synthesis of compounds **39** and **40** via cyclization and transamidation of N-(2-aminoethyl)-2-oxoethanamide derived from (S)-indolylglycinol (Figure 4) [32,33]. We can obtain similar compounds from many sources, including dihydro derivatives and debrominated derivatives [34].

Both compounds **39** and **40** show significant antimicrobial activity against Candida albicans ATCC 44506 and Cryptococcus neoformans ATCC 32045 with MIC value of 1.6–6.2 µg/mL [34]. However, compound **40** shows weak antibacterial activity against Candida albicans and the MIC is 25–100 µg/mL [35]. The activity of compound **39** is pretty strong (MIC, 0.78–3.12 µg/mL), especially against Methicillin-resistant Staphylococcus aureus (MIC, 3.12 µg/mL) and also has potent antifungal activity against Candida albicans (MIC, 6.25 µg/mL) [30,31].

Dragmacidin (**41**) was originally isolated from the deep-water marine sponge of *dragmacidin* sp., and later discovered from *Hexadella* sp. [36]. Along with it, several bis-indole compounds which have similar structure were discovered, including dragmacidin A (**42**) and B (**43**) [37,38]. A year later, Faulkner and co-workers isolated a new alkaloid named dragmacidin C (**44**) from the encrusting gray tunicate Didemnum candidum, which was collected in the southern Gulf of California [39].

Dragmacidin D (**45**) was found in a deep-water marine sponge called *Spongosorites* sp. that was obtained off the coast of southern Australia and had a rotational measurement of +12.5, according to Capon and colleagues [40]. A novel alkaloid known as dragmacidin E (**46**) was discovered in *Spongosorites* sp. that were gathered during a trawling operation off the southern coast of Australia. More recently, a marine sponge of the species *Halicortex* that was obtained in 2000 [41] off the southern coast of Utica Island, Italy, was shown to contain dragmacidin F (**47**)-a novel bioactive bromoindole alkaloid (Figure 9).

Due to their wide range of both biological and pharmacological activities, the necessity of the total synthesis is obvious. Jiang and co-workers reported the total synthesis of compound **41** [42] (Figure 5). In the same year, Cava and co-workers reported a simple synthesis of compound **43** [43]. A short synthetic strategy for various bis-indole marine natural products, including compound **43,** was described by Horne [44]. The first total synthesis of racemic compound **42** via indolyl glycines was accomplished by Kawasaki and co-workers, and the method could be applicable to the syntheses of other members of the dragmacidin family and analogues [45]. In 2005, the facile formal total synthesis of compounds **43** and **44** was reported [46]. The chiral bis-indole alkaloid compound **41** was reported to inhibit in vitro growth of P388 murine leukemia cells (IC_50_ = 15 µg/mL). Additionally, it inhibited the expansion of the cancer cell lines A-549 (human lung), HCT-6 (human colon), and MDAMB (human mammary) (IC50 = 1–10 g/mL) [46]. Serine-threonine protein phosphatase inhibitors have been found as substance **46** and compound **45**, its cometabolite [47].

Nortopsentin A–C (**48**–**50**) (Figure 10) are a new class of compounds discovered in recent years, isolated by Sun and his colleagues from Bahamas’ deep-water sponge, halihondride sponge Spongosorites ruetzleri [11]. Nortopsentin D (**51**) is the simplest bis-indole imidazole alkaloid obtained by catalytic hydrogenation of compounds **48**–**50**, the existence of which has been isolated from Halicondride sponge Spongosorites ruetzleri [11]. Pietra and his colleagues reported a new bis-indole alkaloid named nortopsentin E (**52**), including its separation, structure determination, and biological activity study. The alkaloid is from the deep-water axinellid sponge *Dragmacidon* sp. collected in the south of New Caledonia [48].

In 1994, Ohta and his colleagues reported the synthesis of compound **51** using a continuous and regioselective di-arylation method, based on the treatment of N-protected 2,4,5-tri- and 4,5-di-bromoimidazole derivatives with N-silylated 3-indolylboric acid in the presence of palladium(0) [49]. Using a similar scheme, in 1996, they performed the total synthesis of compounds **48**–**51** (Figure 6, Figure 7 and Figure 8 [30]. Many different synthetic routes have been developed to synthesize these compounds [50,51]. Different from the above scheme, Moody and his colleagues developed a new synthetic route, indole-3 carbonamide prepared from the corresponding amide through thioamide were reacted with 3-bromoacetylindole to obtain 2,4-bis(indolyl)imidazole, N-protected compounds **49** and **51** [48].

Nortopsentins have antifungal activity, and some also have antitumor activity. Compared with their parent compound, their methylated derivatives show a significant increase in the biological activity of P388 [49]. Compounds **48**–**51**, tri- and tetramethylated-nortopsentin B in vitro inhibited P388 murine leukemia cells with IC_50_ of 7.6, 7.8, 1.7, 0.9, and 0.34 µg/mL [11]. In addition to this, in vitro, they inhibited the growth of Bacillus subtilis and Cadida albicans. Compound **50** inhibits the activity of neural nitric oxide synthase and calcineurin, which targets calmodulin, a cofactor shared by the above two enzymes [49]. Activity data show that the brominated compound **49** carrying only one R group is cytotoxic at 0.2 µg/mL, and is more cytotoxic than compound **48** and **50** brominated with two R groups, for P-388 cells, in other words, this change reduces their cytotoxicity to 1.7 µg/mL [30]. Compound **52** is inactive against KB tumor cells, and has almost no antibacterial activity against Staphylococcus aureus. However, after the introduction of methyl groups, although the antibacterial or antifungal activity was not measured, the cytotoxicity towards the KB cell line was highly raised (EC_50_ = 0.014 µg/mL) [48].

Bartik and his colleagues reported on three new bis-indole alkaloids, topsentin (**53**), bromotopsentin (**54**), and deoxytopsentin (**55**), isolated and determined from the Mediterranean sponge Topsentia genitrix collected near Banyuls, France in 1987 [30]. Compound **53** is the first example of brominated bis-indole alkaloids, whose structural feature is to insert a 2-acylimidazole between two indole units substituted or unsubstituted on the benzene ring [52]. After that, Bartik et al. reported the separation of topsentin alkaloids, and Rinehart and colleagues also published a paper that separated compounds **53** and **54** from Caribbean deep-sea sponges and explained their structure. Moreover, they also discovered another new bis-indole alkaloid 4,5-dihydro-6″-deoxybromotopsentin (**56**) [53]. Topsentin C (**57**), a new brominated bisindole alkaloid, was discovered and isolated from the Pacific Ocean sponge *Hexadella* sp. off the coast of British Columbia [38]. The continued interest in these compounds is undoubtedly influenced by their wide range of biological properties. Capon and his colleagues reported on their chemical research on deep-water *Spongosprites* sp. collected on the southern Australian coast. In their paper, they introduced isobromotopsentin (**58**), which had never been reported before [40]. In 1999, four bis-indole alkaloids of topsentins were discovered by Shin’s research team in a sponge Spongosorites genitrix collected from Jaeju island in South Korea, including two new brominated compounds, named bromodeoxytopsentin (**59**) and isobromodeoxytopsentin (**60**) [54] (Figure 11).

In 1987, Braekman and his colleagues reported the first total synthesis of compound **55** [52]. In 1988, Rinhert and his colleagues reported the synthesis of compound **53** [53] (Figure 9). Then, in 2000, under the catalysis of a palladium catalyst, compound **53** was synthesized by cross-coupling reaction at the 5-position of the imidazole ring and acylation at the 2-position [53]. It is believed that the compound **53** molecule is pseudo-symmetric and may be formed by the synthesis of two tryptamine equivalents and the synthon selected in the synthesis is glyoxal indole. If a suitable mixture of indole is selected, it can be condensed with ammonia in equal amounts to obtain the desired asymmetric imidazole (and the other three products), if a single indole is used, a single imidazole is obtained. To confirm the structure of compound **53**, Rinehart and his colleagues reported the synthesis of compound **53**, whose route is completely different from the route reported by Braekman et al. [53]. As shown in Figure 9, the total synthesis of compound **53** is achieved by the condensation of 3-glyoxalylindole and 6-benzyloxy-3-glyoxal in the presence of ammonia, but the synthesis yield is low and non-regioselective [53]. After that, Achab reported a new synthetic method that relies on the continuous introduction of indole to functionalized imidazole derivatives [49].

The family of topsentins compounds exhibits diverse and effective biological activities, such as cytotoxicity, anticancer activity, antifungal activity, antiviral activity, and antibacterial activity. Compound **53** can inhibit the proliferation of cultured human and murine tumor cells at a concentration level of µM (IC_50_ values ranged from 4 to 40 µM) [30]. It shows in vitro activity against P-388 (IC_50_ = 3 µg/mL) and human tumor cell (HCT-8, A-549, T47D: 20 µg/mL) and in vivo activity against P-388 (T/C 137%, 150 mg/kg) and B16 Melanoma (T/C 144%, 37.5 mg/kg) [55]. In 2020, topsentin’s photoprotective properties on UVB-irradiated human epidermal keratinocyte HaCaT cells were discovered by Hwang et al. Topsentin inhibits the expression of COX-2 and the AP-1 and MAPK upstream signaling pathways. Additionally, topsentin blocks the expression of tumor necrosis factor alpha induced protein 2 (TNF-IP2), a target gene for miR-4485, a novel biomarker chosen from a microarray. A model of reconstructive human skin verified topsentin’s photoprotective effects. These results imply that topsentin might be a good cosmetic formulation candidate for skin inflammatory-mediated disorders [56]. Compounds **54** and **58** showed moderate cytotoxicity to the human leukemia cell line K-562 and the IC_50_ of **54** and **58** were 0.6 and 2.1 µg/mL, respectively [57]. In addition to anti-tumor activity, compound **54** also has a very effective local anti-inflammatory activity, being a better inhibitor of phospholipase A2 than manoalide [55]. In addition to the above, compounds **53** and **54** also showed good antiviral activity against HSV-1 vesicular stomatitis virus and the coronavirus A-59.

In 1997, caulersin (**61**) was separated from the alga *Caulerpa serrulata* and became the first member of the bis-indole alkaloid family, with a functionalized seven-membered ring between two indole molecules [58]. Another of these compounds is caulerpin (**62**), a bis-indole whose structure is related to compound **61** isolated from several different green and red algae. It can act as a plant growth regulator and has been shown to inhibit algae growth. It acts in the multixenobiotic resistance (MXR) pump in algae, thereby enhancing the toxicity of xenobiotics [30,59,60,61]. In addition, it can also be used as rust inhibitor in low carbon steel [62] (Figure 12).

The synthesis of compound **61** is carried out in seven steps. The construction of the central seven-membered ring is based on the Michael-type addition of 2,3′-bis(indolyl)-ketone to methylvinyl ketone, followed by intramolecular nucleophilic attack of the resulting 3-oxoalkylation product leading to the substitution of the chlorine atom and ring closure [60] (Figure 10).

In 1977, the synthesis of compound **62** was reported by Maiti et al. It was synthesized from 3-formylindol-2-yl acetic ester with a yield of 5% [61]. Then in 2004, Wahlstrom and his colleagues reported a three-step synthesis of compound **62** [63]. Recently, Mikki and his team reported a four-step synthesis method of compound **61** [64] (Figure 11). Functional analogues of the bisindole alkaloid compound **62** have been prepared by Canche Chay et al. [65] (Figure 12). Using as starting materials 5-substituted indoles, the Vilsmeier Haack reaction with POCl_3_ and DMF forms the corresponding indole aldehyde in good to excellent yields. Subsequent use of dilauroyl peroxide DLP as an oxidative agent which reacts through radical oxidative aromatic substitution of xanthate to produce corresponding malonate derivatives. Further decarboxylation and transesterification reactions between malonate derivatives and NaOMe in MeOH provide monoester indole products. Using piperidine and diethylamine as the base in xylene, the final caulerpin analog can be obtained through the reflux cyclization reaction of monoester indole.

Compounds **61** and **62** have a variety of biological activities. Mao et al. isolated compound **62**, caulerpal A (**63**), and caulerpal B (**64**) from the Chinese green alga *C. taxifolia (Vahl) C. Agardh* and tested their inhibitory activity on hPTP1B [66]. The results show that compound **62** has a strong inhibitory activity on PTP1B, IC_50_ = 3.77 μM, but the inhibitory mechanism has not yet been elucidated.

Caulerpin’s antinociceptive and anti-inflammatory were demonstrated by de Souza et al. Compound **62** significantly inhibited capsaicin induced mouse ear edema by 55.8%, and carrageenan-induced peritonitis by reducing the number of recruit cells by 48.3% [67]. Cavalcante-Silva et al. reported that compound **62** (40 mg/kg) exerts an antinociceptive effect through α_2_-adrenoceptors and 5-HT3 receptors in the writhing test. Therefore, compound **62** is considered to have the prospect of being developed as a dual-acting target analgesic [68].

Liu et al. reported the anti-tumor activity of compound **62** as a cellular hypoxia-targeted [69]. In an experiment based on T47D cells, compound **62** inhibited hypoxia-induced and 1,10-phenanthroline-induced HIF-1 activation. HIF-1 regulates angiogenic factors, including vascular endothelial growth factor (VEGF).

At the same time, in preliminary tests, compound **62** caused a 100% fatality rate on Culex pipiens mosquito larvae at 500 mg/L. As the age of larvae increased, the toxic effect of compound **62** on larvae decreased slightly. This study shows that compound **63** provides potential mosquito control principles that can be used to develop biological control strategies [70].

Canché Chay and his colleagues evaluated compound **62** and its six analogues as an inhibitor of the growth of the Mycobacterium tuberculosis strain, H37Rv. Compound **63** inhibits Mycobacterium by more than 70% and its IC_50_ = 0.24 μM. Besides, its activity is more than twice that of rifampin (IC_50_ = 0.55 μM), which is often used to treat Mycobacterium. Studies have shown that compound **63** is likely to be a potential lead compound for new anti-tuberculosis drugs [65].

In 2013, two new 5-hydroxyindole alkaloids named hyrtinadine B (**65**) and scalaridine A (**66**) were isolated from a Dokdo marine sponge *Scalarispongia* sp. by Lee et al. The cytotoxicity levels of compounds **65** and **66** against human leukemia cells (K562) are IC_50_ = 215.4 μM and 39.5 μM. Compounds **65** and **66**, which are mono- or bis-indoles with heteroaromatic rings, would serve as excellent probes for further research of cancer [71].

Aspertoryadins A–G (**67**–**73**), a group of seven novel quinazoline-containing indole alkaloids, were identified from the marine-derived *Aspergillus* sp. HNMF114 of the bivalve mollusc Sanguinolaria chinensis. Using the techniques previously described, the antibacterial activities of the new compounds against Staphylococcus aureus, Escherichia coli, Bacillus subtilis, and Streptococcus agalactiae as well as the quorum sensing (QS) inhibitory activity against Chromobacterium violaceum CV026 were all assessed. With MIC values of 32 and 32 μg/well, compounds **72** and **73** shown QS inhibitory activity against C. violaceum CV026 [72] (Figure 13).

In 2021, Li et al. discovered one novel pteridine alkaloid, asperpteridinate A, two new prenylated indole alkaloid homodimers, di-6-hydroxydeoxybrevianamide E (**74**) and dinotoamide J (**75**), and eleven recognized compounds from the marine-derived fungus *Aspergillus austroafricanus Y32-2*. Therefore, compound **75** exhibited proangiogenic activity in a PTK787-inducedvascular injury zebrafish model in a dose-dependent manner [73].

Li et al. isolated one new dimeric indole derivative (**76**) from the sponge-derived actinomycete *Rubrobacter radiotolerans*. It exhibited the most effective antichlamydial activity with IC_50_ values of 46.6–96.4 µM in the production of infectious progeny. It appeared to target the mid-stage of the chlamydial developmental cycle by interfering with reticular body replication, but not directly inactivating the infectious elementary body [74].

Antibacterial-guided fractionation of an extract of a deep-water *Topsentia* sp. marine sponge led to the isolation of two new indole alkaloids, tulongicin A (**77**) and dihydrospongotine C (**78**). Compound **77** is the first natural product to contain a di(6-Br-1H-indol-3-yl)methyl group linked to an imidazole core. Both compounds showed strong antimicrobial activity against Staphylococcus aureus (MIC = 1.2 and 3.7 μg/mL) [75] (Figure 14).

Dionemycin (**79**) and 6-OMe-7′,7′′-dichorochromopyrrolic acid (**80**), two novel chlorinated bis-indole alkaloids, were discovered from the deep-sea-derived *Streptomyces* sp. *SCSIO 11791*. Compound **79** demonstrated anti-staphylococcal activity with a MIC range of 1–2 μg/mL against six clinic strains of methicillin-resistant Staphylococcus aureus (MRSA) obtained from human and pig, according to in vitro antibacterial and cytotoxic studies. Furthermore, compound **79** demonstrated cytotoxic action with an IC_50_ range of 3.1–11.2 μM against human cancer cell lines NCI-H460, MDA-MB-231, HCT-116, HepG2, and noncancerous MCF10A [76] (Figure 15).

## 5. Conclusions and Prospect

A valuable source of natural compounds can be found in marine species. Numerous heterocyclic alkaloids and related congeners have been identified and characterized over the past ten years. In recent years, many hetero cyclic alkaloids and congeners with developing value have been derived from them. Most of these indoles have potent cytotoxic activity, and some of them have anti-inflammatory, antifungal, or antiviral activities, making further modification and derivatization of these compounds highly desirable; many derivatives with good activity have been designed and synthesized. In addition, the blank areas of synthetic research on these compounds have been complemented, contributing significantly to their structure–activity studies. Despite the fact that there are more articles about the subject of this review in the literature than ever before, it is reasonable to expect that future research will result in the application of molecules used in clinical treatment.

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
