# Peer review of "Recent Advances of Marine Natural Indole Products in Chemical and Biological Aspects"

_molecules, 2023, doi:10.3390/molecules28052204_

Round 1

Author Response

  • Comment: The English is sometimes a little confusing, and can be improved, may be by a native speaker. 

Response: Thank you very much for your patient review of the manuscript. We carefully checked and revised the manuscript, improved the quality and clarity of the language used in the manuscript. For the issues you mentioned in the PDF file, we all revised and submitted the document for illustrating.

  • Comment: Figures and schemes are not always referred to in the text. Please, make sure that they are cited where appropriate.

Response: Following your comments, we identified these errors and revised the manuscript to ensure they can be cited where appropriate.

Thank you for your valuable comments and patient review for my article, especially for pointing out many details that need to be improved. Thank you for your guidance and efforts.

Reviewer 2 Report

Dear Authors

Do the following steps:

1. Rewrite and redraw the figures. the figures must be organized much better than the current form. obey the same format in figures (for example in some structures you have written Me but in some you have ignored). in some structures you have missed the nitrogen of indole ring.

2. generally, in a review paper you must have a comprehensive conclusion, so please increase the impact and volume of conclusion part.

Author Response

  • Comment: Rewrite and redraw the figures. The figures must be organized much better than the current form. obey the same format in figures (for example in some structures you have written Me but in some you have ignored). in some structures you have missed the nitrogen of indole ring.

Response: We noticed the question and checked all the figures. According to the requirements, we revised the figures and made sure that the same format in figures can be obeyed. The missed nitrogen of indole ring has been added, such as scheme 1, in line 55 and Figure 1, in line 74.
(2) Comment: Generally, in a review paper you must have a comprehensive conclusion, so please increase the impact and volume of conclusion part.

Response: We reviewed the conclusion and found that as your comment, the conclusion is not comprehensive enough. We revised the conclusion, indicating that indoles, due to their excellent biological activities and novel structures, are expected to become an important part of the pharmaceutical field in the future.

Thank you very much for your affirmation and approval to my article.

Reviewer 3 Report

The review title is not appropriate to the contents: indeed, the title is

Recent progress in marine natural indole products and it seems that all marine indole products will be cited and treated and it is not like this.

Several classes are missed. So, please, change the title appropriately indicating which classes of indole derivatives you are reviewing.

If you want to mantain the existing title, an enormous implementation of the review is required to include all marine indole bearing produts.

Author Response

(1) Comment: The review title is not appropriate to the contents: indeed, the title is Rcent progress in marine natural indole products and it seems that all marine indole products will be cited and treated and it is not like this. Several classes are missed. So, please, change the title appropriately indicating which classes of indole derivatives you are reviewing. If you want to maintain the existing title, an enormous implementation of the review is required to include all marine indole bearing products.

Response: Thank you very much for your comments. We agree with your comments and have noticed the inaccuracy of the title. We summarized some marine indole natural products with great pharmacological activity and research value. Besides, we discuss the issues about chemistry, biological evaluation and synthesis. The classees of indole discussed in this review include monomeric indoles, indole peptides, bis-indoles and annelated indoles. Therefore, we consider that it is more accurate to revise the title to “Recent Advances of Marine Natural Indole Products in Chemical and Biological Aspects”.

Round 2

Reviewer 3 Report

Accepted